# Intra-Event and Inter-Event Dependency-Aware Graph Network for Event Argument Extraction

**Hao Li**[1,2], **Yanan Cao**[1,2*], **Yubing Ren**[1,2],
**Fang Fang**[1,2], **Lanxue Zhang**[1,2], **Yingjie Li**[1,2], **Shi Wang**[3]

[1] Institute of Information Engineering, Chinese Academy of Sciences, Beijing, China
[2] School of Cyber Security, University of Chinese Academy of Sciences, Beijing, China
[3] Institute of Computing Technology, Chinese Academy of Sciences, Beijing, China
lihao1998@iie.ac.cn

## Abstract

Event argument extraction is critical to various natural language processing tasks for providing structured information. Existing works usually extract the event arguments one by one, and mostly neglect to build dependency information among event argument roles, especially from the perspective of event structure. Such an approach hinders the model from learning the interactions between different roles. In this paper, we raise our research question: How to adequately model dependencies between different roles for better performance? To this end, we propose an intra-event and inter-event dependency-aware graph network, which uses the event structure as the fundamental unit to construct dependencies between roles. Specifically, we first utilize the dense intra-event graph to construct role dependencies within events, and then construct dependencies between events by retrieving similar events of the current event through the retrieval module. To further optimize dependency information and event representation, we propose a dependency interaction module and two auxiliary tasks to improve the extraction ability of the model in different scenarios. Experimental results on the ACE05, RAMS, and WikiEvents datasets show the great advantages of our proposed approach.

## 1 Introduction

Event Argument Extraction (EAE) aims to identify the arguments of a given event and recognize the specific roles they play, which is one of the important subtasks of Event Extraction (EE). As illustrated by the example in Figure 1, given a *disperseseparate* event triggered by *shipment*, an event argument extractor is expected to identify *Russia*, *arms*, *Armenia* as the event arguments and predict their roles as (Transporter, Origin), Passenger, and Destination, where the *Russia* argument plays the roles of Transporter and Origin respectively.

---
*Corresponding Author.

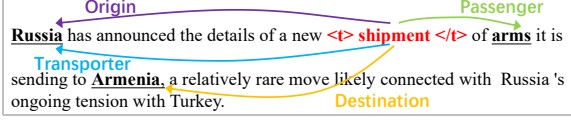
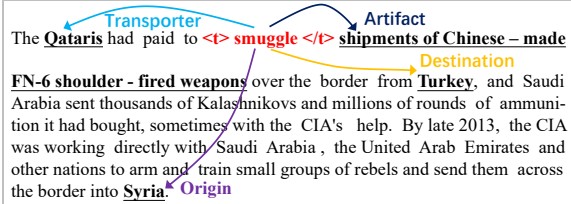

Figure 1: Two examples of event argument extraction from RAMS dataset. Trigger words are included in special tokens <t> and </t>. Underlined words denote arguments and arcs denote roles.

Typical efforts in EAE can be roughly divided into classification-based models (Zhang et al., 2020; Huang and Jia, 2021; Xu et al., 2022; Ma et al., 2022; Ren et al., 2022), Question Answering (QA)-based models (Du and Cardie, 2020; Liu et al., 2020; Li et al., 2020; Wei et al., 2021; Liu et al., 2021), and generation-based models (Li et al., 2021; Lu et al., 2021; Hsu et al., 2022; Ren et al., 2023). In these works, arguments are usually extracted one by one, which mostly neglects to build dependency information among argument roles, especially from the perspective of event structure. For example, classification-based methods typically identify candidate spans and classify their roles independently; QA-based models recognize the boundaries of arguments with role-specific questions; Generation-based methods need to construct role-specific templates for argument filling. Although the state-of-the-art (SOTA) method (Ma et al., 2022) extracts all arguments of an event end-to-end, it still does not explicitly capture dependency among roles.

However, the roles of different arguments within the same event or between similar events clearly

depend on each other. **Firstly**, a complete event usually consists of multiple closely related roles, and establishing dependencies between them can provide intra-event core cues for each other. Furthermore, we count that 11.36% of arguments in RAMS (Ebner et al., 2020) have multiple roles, in which case modeling dependencies is more vital. As shown in Figure 1, the *Russia* plays the roles of Transporter and Origin, respectively. It is intuitive that capturing the dependencies between Transporter and Origin and jointly learning their semantic information can result in mutual gain. **Secondly**, similar events may have similar event structures and share partial roles, so modeling dependencies between roles in similar events can provide inter-event role correlation clues. According to our statistics, 76.98% and 44.00% event types in the ontologies of RAMS and WikiEvents (Li et al., 2021) datasets share more than three roles, respectively. The *disperseseparate* event and *smuggleextract* event in Figure 1 both describe the event of arms shipments between countries, and they share the roles of Transporter, Origin, and Destination, where utilizing the correlation between these two events can be mutually beneficial. Therefore, capturing role dependencies within the same event is essential, and modeling role dependencies between similar events is also crucial.

Built on these motivations, we raise our research question: *How to adequately model dependencies between different roles for better performance?* To this end, we propose an intra-event and inter-**E**vent **D**ependency-aware **G**raph network for **E**AE (EDGE), which uses the event structure as the fundamental unit to build dependencies. Specifically, we first utilize the dense intra-event graph to capture the role dependencies within events. Furthermore, we propose a retrieval module to store and retrieve similar events of the current event, using the retrieved intra-event graph to assist in constructing an inter-event graph for capturing dependencies between events. To propagate the intra-event and inter-event dependency and filter redundant information, we propose a dependency interaction module to fully model the dependency information at different granularities. Finally, to further optimize the event representation, we propose two auxiliary tasks to improve the argument extraction ability of the model in different scenarios.

We summarize our contributions as follows:

- We propose an intra-event and inter-event

dependency-aware graph network for EAE, which can help learn rich role semantic information without using manual label templates.

- When constructing the inter-event graph, we further introduce a retrieval module to retrieve the most similar event to the current event from the memory unit.

- We conduct experiments on three widely used EAE datasets. Through experiments and visual analysis, our model achieves new SOTA due to the dependency modeling, demonstrating its effectiveness.

## 2 Related Works

**Sentence-Level Event Extraction** SEE extracts the event trigger and its arguments from a single sentence. It has achieved great progress in previous studies. Li et al. (2013) employ various hand-designed features to extract events. With the popularity of deep learning, researchers use various neural networks to extract events, such as convolutional neural network (CNN)-based models (Chen et al., 2015; Nguyen and Grishman, 2015), recurrent neural network (RNN)-based models (Nguyen et al., 2016; Sha et al., 2018) and graph neural networks (GNN)-based models (Liu et al., 2018). As pre-trained language models (PLM) have been proven to be powerful in language understanding and generation (Devlin et al., 2019), Yang et al. (2019); Wadden et al. (2019); Du and Cardie (2020); Wang et al. (2021); Lu et al. (2021); Hsu et al. (2022); Liu et al. (2022); Dai et al. (2022) use PLM-based methods to extract events.

**Document-Level Event Extraction** Considering that real-world events are often distributed across sentences, DEE has attracted more attention recently. Different from SEE, DEE extracts the event trigger and its arguments from the whole document. On the task level, most of these works fall into three categories: (1) classification-based models (2) QA-based models (3) generation-based models. Specifically, Zhang et al. (2020); Xu et al. (2021); Huang and Jia (2021); Huang and Peng (2021); Xu et al. (2022); Ma et al. (2022); Ren et al. (2022) employ traditional classification paradigms to identify arguments from text and classify the roles they play in events; Wei et al. (2021); Liu et al. (2021) treat DEE as extractive question answering task, where the extracted arguments are based on carefully constructed natural questions; With the help

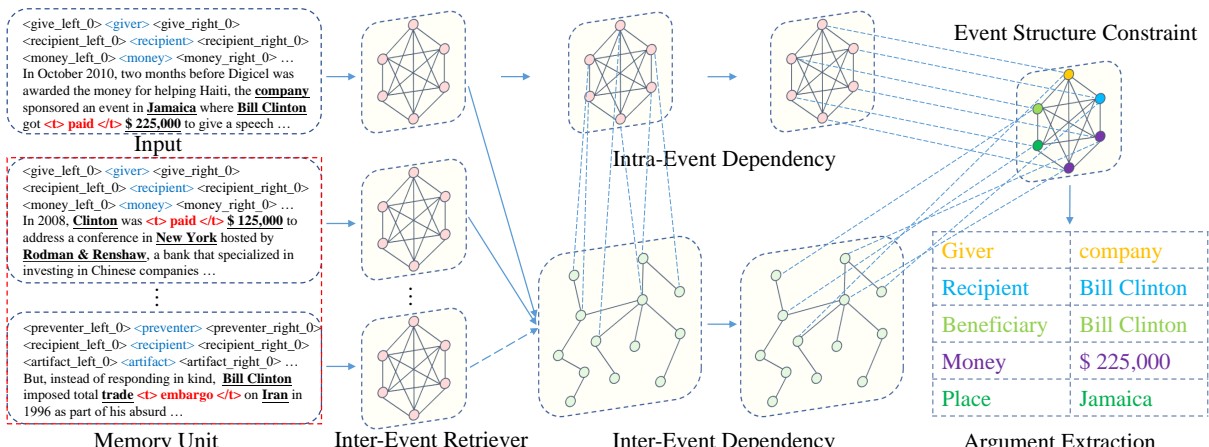

Figure 2: The overall architecture of EDGE. We use a specific example to show the entire process and mark the text description under the specific module. We delineate the real similar and dissimilar events in the memory unit. Furthermore, colors distinguish different argument roles, solid lines with arrows indicate the flow of data, and dashed lines indicate dependency interactions. Finally, the specific extraction results of this example are described.

of the pre-trained Encoder-Decoder Transformer architecture (Lewis et al., 2020), Li et al. (2021); Du et al. (2022); Ren et al. (2023) convert the extraction task to a sequence-to-sequence generation task, and get the arguments and corresponding roles in the event in a generated way.

## 3 Methodology

In this section, we first formulate the EAE task, then introduce the overall framework of our model, as illustrated in Figure 2, which contains four core components: Role-Aware Encoding, Bi-Granularity Dependency Construction, Intra2Inter Dependency Interaction, and Dependency-Aware Argument Extraction.

**Task Definition** We first describe the task formalization of EAE. Formally, given a context $X = \{x_1, ..., x_n\}$ of $n$ words, the event trigger word $x^t \in X$, the event type $t \in \mathcal{T}$, where $\mathcal{T} = \{t_1, ..., t_u\}$ is the set of all $u$ different event types and the set of event-specific role types $\mathcal{R}^t$. The EAE task aims to extract all the arguments $\{a_1, ..., a_d\}$ related to $x^t$ in $X$, and assign a role $r \in \mathcal{R}^t$ to each extracted argument $a_i$, where $a_i$ is a text span in $X$.

### 3.1 Role-Aware Encoding

Given context $X$ and the set of event-specific role types $\mathcal{R}^t$, this module generates event-oriented context representation and context-oriented role representation. We use pre-trained BART (Lewis et al., 2020) as our encoding model, a standard

Transformer-based pre-trained language model consisting of both an Encoder and a Decoder.

Following previous works (Ma et al., 2022), we feed the context into BART-Encoder and the role into BART-Decoder separately. Specifically, given the current trigger word $x^t$, we first define the trigger markers $\langle \mathbf{t} \rangle$ and $\langle \mathbf{/t} \rangle$ as special tokens and insert them into context $X$ before and after the trigger word $x^t$ to create an input context $\tilde{X}$. In the next step, the BART-Encoder is employed to encode the input context $\tilde{X}$ to obtain the sequence representation $H_x^{enc}$. Finally, benefiting from mutual interaction information at the cross-attention layers in the decoder module, the BART-Decoder is applied to learn richer representations for the context and roles, returning $H_x = \text{BART-Decoder}(H_x^{enc}; H_x^{enc})$ and $H_r = \text{BART-Decoder}(\tilde{\mathcal{R}}^t; H_x^{enc})$ for the context and roles, where $\tilde{\mathcal{R}}^t$ denotes connecting different roles $r_i$ with learnable, role-specific pseudo tokens, $H_x$ denotes the event-oriented context representation, $H_r = \{h_{r_1}, ..., h_{r_d}\}$ denotes context-oriented role representation, and $d$ denotes the maximum number of roles for this event[1].

### 3.2 Bi-Granularity Dependency Construction

The dependency construction module constructs two granularities of dependencies: intra-event dependency and inter-event dependency.

**Intra-Event Dependency** Considering the one-to-many form of trigger words and arguments in the

---
[1]The number of each role is determined by the maximum number of arguments for this role in the training set.

event structure, the roles participating in the same event are closely related, so the intra-event graph representing the dependency within the event is a fully connected graph. For each event type $t$, the nodes $v_i \in V_l$ in the event are context-oriented role representation $h_{r_i} \in H_r$, using the co-occurrence information of roles of event type $t$ in the training set as dependency weight:

$$\alpha_{ij}^t = \frac{\text{count}(t, r_i, r_j)}{\text{count}(t, r_i)} \quad (1)$$

Finally, after normalized symmetric (Chung, 1997) $\tilde{A} = D^{-\frac{1}{2}} A D^{-\frac{1}{2}}$, where $D$ denotes degree matrix. The intra-event graph with weight $\tilde{\alpha}_{ij}^t$ is used to enhance the representations for $h_{r_i}$ via a Graph Convolutional Network (GCN) (Kipf and Welling, 2017) with $K$ layers:

$$z_i^k = \text{ReLU}\left( \sum_{v_j \in V_l} \tilde{\alpha}_{ij}^t z_j^{k-1} w^k + b^k \right) \quad (2)$$

where $w^k$ and $b^k$ are learnable parameters, and $z_i^k$ represents the role representation for $v_i$ at the $k$-th layer of GCN ($z_i^0 = h_{r_i}$).

**Inter-Event Dependency**  Different from intra-event dependencies, inter-event dependencies are represented by the inter-event graph to construct role dependency information between different events. In order to model the role dependency information between different events, we propose a memory unit $M$ to store all the intra-event graphs in the form of tensors, which are dynamically updated according to the corresponding indexes at each training epoch. The intra-event graph obtained by the retrieval module will be used to construct the inter-event graph, and the dependencies between different events will be constructed. Specifically, we first use cosine similarity in the retrieval module to retrieve the intra-event graph of the event most similar to the current event:

$$\text{S}(m_i | m_c) = \frac{f(H_{r_i}) \cdot f(H_{r_c})}{||f(H_{r_i})|| \times ||f(H_{r_c})||}$$
$$m_s = \underset{m_i \in M}{\arg\max}(\text{S}(m_i | m_c)) \quad (3)$$

where $f$ represents the mean pooling operation, $m_c$ and $m_s$ denote the current event and the most similar event, respectively. $m_i \in M$ denotes the event in the memory unit, and the corresponding $H_{r_c}$ and $H_{r_i}$ represent the context-oriented role representation of the current event and the event in the memory unit respectively.

Afterward, we use the intra-event graphs of the current event $m_c$ and the most similar event $m_s$ to construct the inter-event graph. Specifically, the nodes $V_g$ of the inter-event graph are the roles of multiple event types, which are obtained by mapping the corresponding nodes in the intra-event graphs of the current event and the most similar event. The dependency weights of the inter-event graph are constructed from the training set based on the co-occurrence information of all roles:

$$\beta_{ij} = \frac{\text{count}(r_i, r_j)}{\text{count}(r_i)} \quad (4)$$

Different from the intra-event graph, the inter-event graph captures dependencies between events, so its nodes and edges correspond to multiple event types and roles, not just a single event type. Finally, the representation of the inter-event graph is learned using a GCN similar to Equation 2.

### 3.3 Intra2Inter Dependency Interaction

We use two different GCNs to learn the representation of the intra-event graph and the inter-event graph. To propagate dependency information at different granularities, we align nodes representing the same roles in the intra-event and inter-event graph at each layer of the GCN: $Z_l^k = Z_g^k = (Z_l^k + Z_g^k)/2$, where $Z_l^k$ and $Z_g^k$ represent the node representation of the intra-event and inter-event graph at the $k$-th layer of the GCN, respectively. The dependency representations are obtained by the above two $K$-layer GCNs and $K$ times of dependency interaction, and finally the dependency information is assigned to the context representations by concatenating: $H_r' = \text{FFN}(\text{concat}(H_r, Z_l^K))$, where FFN is a feed-forward network.

**Event Structure Constraint**  To learn and model the event structure information, we add two additional event structure constraints: the role constraint, which uses the nodes of the dependency graph to predict the role type, and the event constraint, which uses the entire dependency graph to predict the event type:

$$\mathcal{L}_r = \sum_{X \in \mathcal{D}} \sum_{h_{r_i}' \in H_r'} -\log(\text{Softmax}(h_{r_i}' w^r))$$
$$\mathcal{L}_e = \sum_{X \in \mathcal{D}} -\log(\text{Softmax}(f(H_r') w^e)) \quad (5)$$

where $w^r$ and $w^e$ are learnable parameters, $\mathcal{L}_r$ and $\mathcal{L}_e$ represent role constraint loss and event constraint loss respectively, and $\mathcal{D}$ ranges over all context in the dataset.

## 3.4 Dependency-Aware Argument Extraction

Given the context representation $H_x$ and the set of dependency-aware role representation $H'_r$, each $h'_{r_k}$ aims to extract the corresponding argument span $(s_k, e_k)$ from the context $X$, where $s_k$ and $e_k$ are the start and end word indices of the argument. The span $(s_k, e_k)$ will be set to $(0, 0)$ if $h'_{r_k}$ has no corresponding argument (the current event has no argument about the role, or the argument-slot number of this role exceeds the actual argument number). For each $h'_{r_k}$, we calculate the distribution of each token being selected as the start and end position of the argument (Du and Cardie, 2020; Ma et al., 2022):

$$\text{logit}_k^{start} = (h'_{r_k} \odot w^{start})H_x \in \mathbb{R}^n$$
$$\text{logit}_k^{end} = (h'_{r_k} \odot w^{end})H_x \in \mathbb{R}^n \qquad (6)$$

where $w^{start}$ and $w^{end}$ are learnable parameters, and $\odot$ represents the element-wise multiplication.

We calculate probabilities where the start and end positions are located as follows:

$$p_k^{start} = \text{Softmax}(\text{logit}_k^{start}) \in \mathbb{R}^n$$
$$p_k^{end} = \text{Softmax}(\text{logit}_k^{end}) \in \mathbb{R}^n \qquad (7)$$

The argument extraction loss is defined as:

$$\mathcal{L}_k(X) = -(\log p_k^{start}(s_k) + \log p_k^{end}(e_k))$$
$$\mathcal{L}_a = \sum_{X \in \mathcal{D}} \sum_{k=1}^{d} \mathcal{L}_k(X) \qquad (8)$$

The overall loss function is divided into three parts: argument extraction loss $\mathcal{L}_a$, role constraint loss $\mathcal{L}_r$, and event constraint loss $\mathcal{L}_e$:

$$\mathcal{L} = \mathcal{L}_a + \lambda_1 \mathcal{L}_r + \lambda_2 \mathcal{L}_e \qquad (9)$$

and $\lambda_1$, $\lambda_2$ are hyper-parameters.

## 3.5 Inference

During inference, we consider all candidate spans of the argument as $\mathcal{C} = \{(i,j)|(i,j) \in n^2, 0 < j - i \leq \delta\} \cup \{(0,0)\}$, ensuring that the length of all spans does not exceed the threshold $\delta$ and $(0, 0)$ means there is no argument. The argument of each $h'_{r_k}$ is extracted by enumerating and scoring all candidate spans as:

$$\text{score}_k(i, j) = \text{logit}_k^{start}(i) + \text{logit}_k^{end}(j) \qquad (10)$$

The span with the highest score will be selected as the prediction result. Specifically, the prediction span of $h'_{r_k}$ is computed:

$$(\hat{s}_k, \hat{e}_k) = \underset{(i,j) \in \mathcal{C}}{\arg\max} \, \text{score}_k(i, j) \qquad (11)$$

# 4 Experiments

## 4.1 Experimental Setup

**Datasets** We conduct experiments on three commonly used EAE datasets: RAMS (Ebner et al., 2020), WikiEvents (Li et al., 2021) and ACE05 (Doddington et al., 2004). RAMS and WikiEvents are widely used document-level EAE datasets, and ACE05 is another classic dataset for sentence-level EAE tasks. Detailed statistics are listed in Appendix A.

**Evaluation Metrics** We measure the performance with three evaluation metrics: (1) **Argument Identification** (Arg-I): an event argument is correctly identified if its offsets and event type match those of any of the argument mentions. (2) **Argument Classification** (Arg-C): an event argument is correctly classified if its role type is also correct. (3) For WikiEvents dataset, we follow previous work (Li et al., 2021), additionally evaluate **Argument Head Classification** (Head-C), which only concerns the matching of the headword of an argument. The F-measure (F1) score is used to evaluate the performance of the model.

**Implementation Details** We initialize our models with the pre-trained BART base models (Lewis et al., 2020). For important hyper-parameters and details, please refer to Appendix B.

**Baselines** We compare our model to several representative and competitive baselines:

- EEQA (Du and Cardie, 2020) treats sentence-level EAE as a QA task, and obtains the start and end offsets of the argument spans through question answering.

- FEAE (Wei et al., 2021) is a QA-based method extended to EAE by considering argument interactions via knowledge distillation.

- DocMRC (Liu et al., 2021) is another QA-based method, assisted by two data augmentation regimes.

- BART-Gen (Li et al., 2021) defines EAE as a sequence-to-sequence task and generates corresponding arguments in a predefined format.

- PAIE (Ma et al., 2022) is a prompt-based method. It extracts argument spans by constructing role templates and is the current SOTA model. For a fair comparison with our model, we use its soft prompt setting.

| Model | PLM | ACE05 | | RAMS | | WikiEvents | | |
|---|---|---|---|---|---|---|---|---|
| | | Arg-I | Arg-C | Arg-I | Arg-C | Arg-I | Arg-C | Head-C |
| FEAE (Wei et al., 2021) | BERT | - | - | 53.5 | 47.4 | - | - | - |
| DocMRC (Liu et al., 2021) | BERT | - | - | - | 45.7 | - | 43.3 | - |
| EEQA (Du and Cardie, 2020)† | BERT | 68.2 | 65.4 | 46.4 | 44.0 | 54.3 | 53.2 | 56.9 |
| BART-Gen (Li et al., 2021)† | BART | 59.6 | 55.0 | 50.9 | 44.9 | 47.5 | 41.7 | 44.2 |
| EEQA-BART (Ma et al., 2022)† | BART | 69.6 | 67.7 | 49.4 | 46.3 | 60.3 | 57.1 | 61.4 |
| PAIE (Ma et al., 2022) | BART | 74.3 | 70.1 | 53.9 | 48.0 | 66.2 | 60.1 | 63.8 |
| EDGE (Ours) | BART | **75.3** | **70.6** | **55.2** | **49.7** | **68.2** | **62.8** | **65.9** |

Table 1: Event argument extraction results on three mainstream datasets. Both BERT and BART here are base-scale. Result marked † are from (Ma et al., 2022). The best results are marked in bold font.

| Model | Arg-I | Arg-C |
|---|---|---|
| EDGE | **55.2** | **49.7** |
| w/o inter-event dependency | 54.0 | 47.8 |
| w/o retrieval module | 54.7 | 48.5 |
| w/o role constraint | 55.1 | 48.9 |
| w/o event constraint | 55.0 | 49.4 |

Table 2: Ablation study results on RAMS.

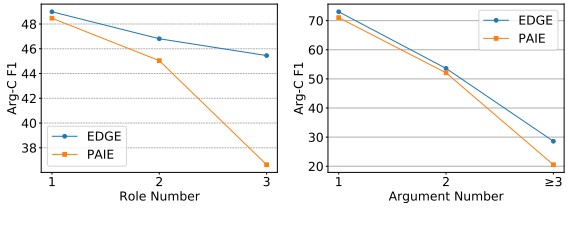

(a) Same-Argument Role   (b) Same-Role Argument

Figure 3: Experimental results of argument-role assignment. (a). Different numbers of roles for an argument on RAMS. (b). Different numbers of arguments for a role on WikiEvents.

## 4.2 Overall Performance

Table 1 compares our method with all baselines on the ACE05, RAMS, and WikiEvents datasets. We have several observations and discussions:

(1) Our method consistently outperforms the latest baselines and achieves the SOTA performance. Specifically, our method achieves 75.3 Arg-I F1 and 70.6 Arg-C F1 on ACE05. On the RAMS dataset, Our method improves 1.3~8.8 Arg-I F1 and 1.7~5.7 Arg-C F1 over previous methods. On the WikiEvents dataset, our method improves 2.0~20.7 Arg-I F1, 2.7~21.1 Arg-C F1 and 2.1~21.7 Head-C F1. These results demonstrate that our method is superior to previous methods in both argument identification and classification.

(2) We find that extracting all arguments from an event end-to-end is more efficient than extracting arguments one at a time. Compared with EEQA, the end-to-end methods (our method and PAIE) achieve different degrees of performance improvement, which demonstrates the existence of rich implicit dependency information between roles. Independent role-specific QA methods can only extract one argument at a time, ignore this dependency information, and still struggle with the task of argument identification and classification.

(3) Compared to the current SOTA model PAIE,

our model not only achieves better results on all evaluation metrics but also achieves higher performance gains on the more difficult Arg-C metric of document-level EAE. This result shows that simply implicitly constructing dependencies between argument roles is not sufficient for EAE. By explicitly constructing dependencies between roles, our model can more accurately identify and extract the corresponding arguments.

## 4.3 Ablation Study

We conduct ablation studies to investigate the effectiveness of each component of our approach and the results are shown in Table 2. Specifically, "w/o inter-event dependency" means removing inter-event dependencies to quantify its effectiveness, "w/o retrieval module" means removing the retrieval module and using only current events for graph representation, "w/o role constraint" and "w/o event constraint" mean that the two auxiliary constraints are not considered respectively.

We can find that each module can help boost the performance of EAE, removing any module will lead to different degrees of performance degrada-

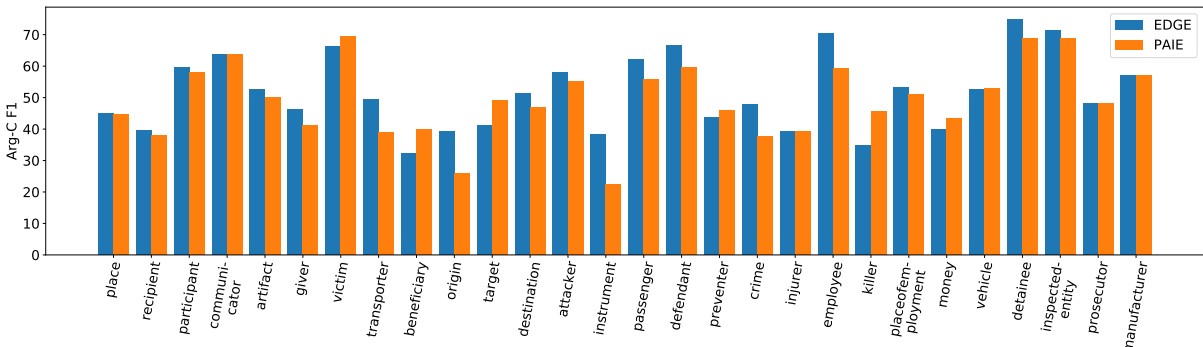

Figure 4: Experimental results for each argument role on RAMS (the argument number > 10). The ordering of roles is from the largest to the smallest number of corresponding arguments.

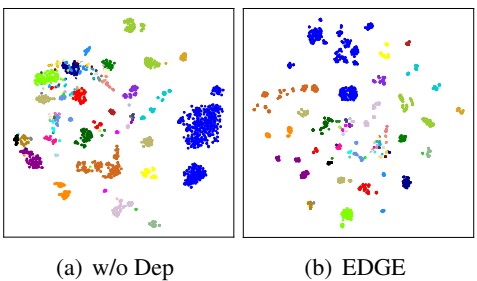

(a) w/o Dep      (b) EDGE

Figure 5: 2D visualization of projected role representation. (a). Role representation trained by w/o Dep (EDGE without any role dependency). (b). Role representation trained by EDGE.

tion. To be specific, **(1)** Without inter-event dependency, the performance drops dramatically by about 1.9 points in terms of Arg-C F1. This justifies that it is necessary to build dependencies between events and establish connections between different events. **(2)** Removing the retrieval module has a performance drop of 1.2 Arg-C F1, which indicates that our retrieval module can improve task performance by efficiently retrieving similar events. **(3)** Similar observations can be found in other modules, removing the role constraint and event constraint results in different degrees of performance degradation, indicating that the event structure is helpful for argument identification and classification.

## 5 Analysis on Real Scenario

### 5.1 Same-Argument Role Assignment

In the EAE task, a complete event usually includes multiple arguments and corresponding roles. Furthermore, an argument may play more than one role in an event, and extracting these arguments is more important for event understanding, but also more difficult. In order to explore the performance of our

model in this scenario, we split the data according to the number of roles corresponding to the argument. Figure 3(a) shows the experimental results in scenarios with different numbers of roles. Compared with the baseline, our method achieves better results in all scenarios and exhibits better robustness. It is worth noting that our method achieves greater performance improvement as the number of roles corresponding to arguments increases, which indicates the necessity of building dependencies among the roles in such scenarios.

### 5.2 Same-Role Argument Assignment

In addition to one argument playing multiple roles, there may be multiple arguments playing the same role in a complex event. These arguments are indispensable in the event, so it is important to extract them to understand the complex event. In order to explore the performance of our method in this scenario, we split the data according to the number of arguments corresponding to the role, and the experimental results are shown in Figure 3(b). Our method achieves better performance than the baseline in all scenarios, especially in the most complex scenarios where the number of arguments is no less than three, achieving 8.0 Arg-C F1 gains. This demonstrates that our method can alleviate the argument and role assignment and more accurately extract multiple arguments for the same role by building intra-event and inter-event dependencies to learn semantic information between roles.

### 5.3 Role Performance Analysis

To examine the performance on the role of the long-tail distribution (Ebner et al., 2020), we calculate the Arg-C F1 of the method on each role. The experimental results are shown in Figure 4. Compared with the PAIE, our method achieves the same

| | | | | |
|---|---|---|---|---|
| Input | Damascus has reacted harshly to the **\<t\> bombing**Instument **\</t\>** of **Kurdish militias**Target in **northern Syria**Place on Thursday morning by **Turkey 's air force**Attacker, vowing to intervene next time Ankara sends its planes over its border. Read more In a statement, the Syrian Defense Ministry accused Turkey of "flagrant aggression, which targeted innocent citizens," saying that it considers it " a dangerous development that could escalate the situation." | | | |
| w/o Dep | **Attacker** | **Target** | **Instrument** | **Place** |
| | Turkey 's air force √ | Kurdish militias √ | air force × | northern Syria √ |
| EDGE | Ukraine authorities said Moutaux' aimed to blow up a Muslim mosque, a Jewish synagogue, tax collection organisations, police patrol units and numerous other locations' French regional newspaper L'Est Republicain identified the man as Gregoire Moutaux and said investigators raided his home (pictured) in Nant-le-Petit near the eastern city of Nancy in late May. It managed to murder 130 people, with **suicide bombers**Attacker **\<t\> exploding \</t\>** their **devices**Instrument around the **Stade de France**Target during a **football friendly**Place between France and Germany… | | | |
| | **Attacker** | **Target** | **Instrument** | **Place** |
| | Turkey 's air force √ | Kurdish militias √ | bombing √ | northern Syria √ |

Figure 6: The case study of our proposed EDGE and w/o Dep. The event triggers are included in special tokens \<t\> and \</t\>, and the event arguments are shown in underline, with their roles in subscripts. In our method EDGE, the most similar event obtained by the retrieval module is additionally shown.

or better results on 71.43% (20/28) roles. Specifically, our method achieves a performance improvement of more than 10 Arg-C F1 on multiple roles, such as instrument (15.79), origin (13.17), and employee (11.11). All these observations verify that our method significantly improves the extraction performance of different argument roles by constructing intra-event and inter-event role dependencies, which helps alleviate the long-tail problem.

## 5.4 Visualization Analysis

In order to explore whether our method fully models the semantic information of argument roles, we map all roles on the RAMS development set to the 2-dimensional vector space through the t-Distributed Stochastic Neighbor Embedding (t-SNE) (Van der Maaten and Hinton, 2008) method for visual analysis. The visualization results are shown in Figure 5, where different roles are displayed in different colors. Compared with w/o Dep, our method can cluster the roles of similar semantics together according to the semantic information of the context, showing better intra-role and inter-role distance under different events. This indicates that by constructing dependencies between argument roles and event structure constraints, our method can better learn the semantic information of roles in different events.

## 5.5 Case Study

In this subsection, we show an example from RAMS dataset to illustrate the capability of our method. The example is presented in Figure 6, including the input document, event argument extraction results, and the most similar event retrieved

from EDGE. Specifically, EDGE can accurately extract all the arguments, while w/o Dep fails. At the same time, the input documents and retrieved events are armed attacks that cause citizens to be injured, indicating that our retrieval module can accurately retrieve similar events and construct the inter-event role dependencies between these events. The results show that our method can retrieve similar events with similar semantics and event structure as the input document, and all arguments in events can be accurately extracted by constructing role dependencies between events.

## 6 Conclusion

In this paper, we propose a novel method based on role dependencies for EAE. Specifically, we first utilize the intra-event graph and the inter-event graph retrieved from the intra-event graph to automatically construct intra-event and inter-event dependencies, and then fully learn and model the dependencies of roles in different scenarios through two different GCNs. To further optimize dependency information and event representation, we propose a dependency interaction module and two auxiliary tasks to improve the argument extraction ability of the model in different scenarios. We conduct extensive experiments on three EAE datasets and compare our model with several strong baselines. The results show that our model achieves SOTA performance and outperforms all baselines in different scenarios. In further work, we are interested in constructing dependency information for other information extraction tasks and would like to apply our model to these tasks, such as named entity recognition and relation extraction.

## Limitations

We discuss the limitations of our research as follows:

- Our method uses additional graph convolutional networks to construct the dependency of argument roles, resulting in an increase in training parameters, and one training process occupies nearly one NVIDIA V100 32GB GPU.

- Our method requires event ontology information to construct intra-event and inter-event graphs, and is not suitable for scenarios where event ontology is unknown.

## Acknowledgements

This work is supported by the National Key Research and Development Program of China (NO.2022YFB3102200) and Strategic Priority Research Program of the Chinese Academy of Sciences with No. XDC02030400.

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

# A    Dataset statistics

RAMS focuses on a document-level EAE task, including 139 event types and 65 argument roles, with more than 9k events. WikiEvents is another widely used document-level EAE dataset, which contains 50 event types and 59 argument roles, containing more than 3.9k events. ACE05 is a sentence-level extraction dataset, including 33 event types and 35 argument roles, with more than 5k events. We follow the official data split of each dataset. For ACE05, we follow the pre-processing procedure of DyGIE++ (Wadden et al., 2019). The specific statistics of the datasets are listed in Table 3 and 4.

| Dataset | #Split | #Doc | #Event | #Argument |
|---|---|---|---|---|
| RAMS | Train | 3,194 | 7,329 | 17,026 |
| | Dev | 399 | 924 | 2,188 |
| | Test | 400 | 871 | 2,023 |
| WikiEvents | Train | 206 | 3,241 | 4,542 |
| | Dev | 20 | 345 | 428 |
| | Test | 20 | 365 | 566 |

Table 3: Data statistics of RAMS and WikiEvents.

| Dataset | #Split | #Sent | #Event | #Argument |
|---|---|---|---|---|
| ACE05 | Train | 17,172 | 4,202 | 4,859 |
| | Dev | 923 | 450 | 605 |
| | Test | 832 | 403 | 576 |

Table 4: Data statistics of ACE05.

# B    Implementation Details

In our implementations, we use the HuggingFace's Transformers library to implement the BART base model. The model is optimized with the AdamW optimizer (Loshchilov and Hutter, 2019). Other important hyper-parameters are shown in Table 5.

| Hyper-parameter | Value |
|---|---|
| Training epochs | 10 |
| Learning rate | 2e-5 (RAMS) / 3e-5 (Other) |
| Layers of GCN | 2 |
| Max span length | 8 |
| Window size | 250 |
| Max encoder seq length | 500 / 192 (ACE05) |
| Max decoder seq length | 80 |
| $\lambda_1$ | 0.2 |
| $\lambda_2$ | 0.5 |

Table 5: Hyper-parameters for EDGE.