# OpenReview forum: "Intra-Event and Inter-Event Dependency-Aware Graph Network for Event Argument Extraction"
_EMNLP/2023/Conference — EMNLP 2023 Findings_

### Official Review · Reviewer_C7eU · 2023-08-03

**Typos Grammar Style And Presentation Improvements:** See reasons to reject.
**Soundness:** 2

**Excitement:**

3: Ambivalent: It has merits (e.g., it reports state-of-the-art results, the idea is nice), but there are key weaknesses (e.g., it describes incremental work), and it can significantly benefit from another round of revision. However, I won't object to accepting it if my co-reviewers champion it.

**Missing References:**

See reasons to reject.

**Paper Topic And Main Contributions:**

This work presents an event argument extraction model that models the intra and inter-event argument dependency with graph neural networks. This work argues that different arguments within an event or between similar events are highly depended, and the dependency can be modeled by the proposed architecture. The authors conduct experiments on three datasets, ACE05, RAMS, and WikiEvents, and find that their proposed methods can outperform several baselines.

**Questions For The Authors:**

See reasons to reject.

**Reasons To Accept:**

1. The motivation for using GCN to model the argument dependency is clear.

2. The experimental results show promising state-of-the-art performance, indicating the effectiveness of their proposed method.

**Reasons To Reject:**

1. My major concern with this paper is the unclear writing. There are many technical details that are not explained properly, making it difficult to understand the architecture and reproduce this work. Below are a few examples:

- In Line 207, the decoder input parameters are not quite confusing. The first parameter can be a list of vectors ($H_x = Decoder(H_X^{enc}; H_X^{enc})$ or a list of tokens ($H_r = Decoder( \hat{\mathcal{R}}^t, H_x^{(enc)})$). It is also not clear how these two parameters are associated with the architecture of BART decoder.
- In the description of intra-event dependency (starting from Line 216), they introduce an intra-event graph structure related to each event type, and the graph is initialized with role representations $H_r$. However, in Line 207, we know that $H_r$ is not only a role-dependent representation, but is also trigger-depended ($\hat{X}$ introduces special tags, $\langle t \rangle$ $\langle /t \rangle$ for trigger). And in the description in Line 197, we know that there is only one trigger word marked in $\hat{X}$. Therefore, it is unclear how to choose or compute $H_r$ when there are multiple triggers (not co-referential) that have the same event type.
- In Line 245, they mention that all intra-event graphs are stored and updated at each training epoch. But as in the previous discussion, intra-graph representations are contextual-dependent. How to store those representations and applied to unseen contextual is also unclear.
- In Line 253, $H_{r_i}$ and $H_{r_c}$ are introduced without further clarification.

I would strongly suggest the authors carefully reorganize and polish the descriptions in this paper.

2. Another concern is that this paper claims that previous work "neglects to build dependency information among argument roles". But there is indeed some work discussing the dependencies. One paper I can remember is [Lin et al., 2020]. Probably there are some older ones with integer linear programming. I would recommend the authors make clear comparisons and discussions on this line of research.

**Reference**

[Lin et al., 2020] A Joint Neural Model for Information Extraction with Global Features

**Reproducibility:**

2: Would be hard pressed to reproduce the results. The contribution depends on data that are simply not available outside the author's institution or consortium; not enough details are provided.

**Reviewer Confidence:**

3: Pretty sure, but there's a chance I missed something. Although I have a good feel for this area in general, I did not carefully check the paper's details, e.g., the math, experimental design, or novelty.

---

> ### Author Rebuttal · Authors · 2023-08-27
>
> We thank all reviewers for their time and for giving valuable and supportive comments. We will take these comments into consideration and make the necessary revisions.
>
> Question 1: It is also not clear how these two parameters are associated with the architecture of BART decoder.
>
> The two parameters of the Decoder represent input ids and encoder hidden states respectively. In line 207, $H_x^{(enc)} = {Encoder}(\tilde{X})$ and $H_x = {Decoder}(H_x^{(enc)};H_x^{(enc)})$ represent a completed Encoder-Decoder process, and there is no need to repeatedly encode the input ids, so the two parameters of Decoder in $H_x = {Decoder}(H_x^{(enc)};H_x^{(enc)})$ are both $H_x^{(enc)}$. For $H_{r} = {Decoder}(\tilde{\mathcal{R}}^{t}; H_x^{(enc)})$, it represents a separate decoding process, so the two parameters are $\tilde{\mathcal{R}}^{t}$ and $H_x^{(enc)}$ respectively.
>
> Question 2: How to choose or compute $H_r$ when there are multiple triggers?
>
> For the event argument extraction task, the trigger word and event type in the current context will be given. Following the classic work (Ma et al., 2022), we segment the context according to the window centered on the trigger word (the window length is set in the appendix, consistent with previous work), which not only guarantees the input length limit of the pre-trained model, but also divides a small number of contexts with multiple trigger words into a single event, which reduces the difficulty of extraction and improves task performance.
>
> Question 3: How to store those representations and applied to unseen contextual is also unclear.
>
> We propose a memory unit $M$ to store all intra-event graphs during the training phase. Specifically, we save the id of the intra-event graphs and store the intra-event graphs in the form of tensors (gradients removed). In each training epoch, we will lookup the saved tensors by id and update it.
>
> Question 4: $H_{r_i}$ and $H_{r_c}$ are introduced without further clarification.
>
> We explained in line 256 that $m_c$ represents the current event, $m_i \in M$  represents the event in the memory unit, and the corresponding $H_{r_c}$ and $H_{r_i}$ represent the context-oriented role representation of the current event and the event in the memory unit respectively.
>
> Question 5: Previous work "neglects to build dependency information among argument roles", clear comparisons and discussions on this line of research.
>
> We use "mostly" and "usually" in the paper. In event-related tasks (not just event argument extraction), previous work (e.g. (Lin et al., 2020; Cui et al., 2020; Lv et al., 2021; Xu et al., 2021)) mostly builds dependencies between entities, instances, or sentences. However, we innovatively use the event structure as a fundamental unit to build the dependencies of the argument roles. Our advantage is that it conforms to the definition of the event structure and does not require a lot of pre-processing work. At the same time, taking the event as a unit can also help us retrieve meaningful similar events and complete the argument extraction of the current event.
>
> Reference
>
> [Lin et al., 2020] A Joint Neural Model for Information Extraction with Global Features
>
> They explicitly model cross-subtask and cross-instance inter-dependencies.
>
> [Cui et al., 2020] Edge-Enhanced Graph Convolution Networks for
> Event Detection with Syntactic Relation
>
> They integrate syntactic structure and typed dependency labels to improve neural event detection.
>
> [Lv et al., 2021] HGEED: Hierarchical graph enhanced event detection
>
> They aim to use the information between sentences in the document to build dependencies.
>
> [Xu et al., 2021] Document-level Event Extraction via Heterogeneous Graph-based Interaction Model with a Tracker
>
> They focus on the global interactions among sentences and entity mentions.

---

### Official Review · Reviewer_Y77w · 2023-08-04

**Soundness:** 3

**Excitement:**

3: Ambivalent: It has merits (e.g., it reports state-of-the-art results, the idea is nice), but there are key weaknesses (e.g., it describes incremental work), and it can significantly benefit from another round of revision. However, I won't object to accepting it if my co-reviewers champion it.

**Paper Topic And Main Contributions:**

 The authors propose an interesting approach to modeling intra-event and inter-event dependencies for event argument extraction. The idea of constructing graphical networks to capture role relationships within and across events is novel and intuitively appealing.

**Reasons To Accept:**

Pos:

* The idea of constructing intra-event and inter-event dependency graphs is creative, intuitive, and well-motivated. Encoding structured knowledge about event roles into graphical networks is an elegant approach.

* Incorporating an event memory and retrieval module is innovative, allowing inter-event dependencies to be dynamically constructed at inference time.

**Reasons To Reject:**

Neg:

* The performance gains over prior methods appear relatively small based on the results presented.

* The authors make a compelling case that long-range trigger-argument dependencies are captured, but the results in Table 3 are not fully convincing.

* The results should reflect the variance of the scores from multiple experiments with different seeds.


**Reproducibility:**

4: Could mostly reproduce the results, but there may be some variation because of sample variance or minor variations in their interpretation of the protocol or method.

**Reviewer Confidence:**

4: Quite sure. I tried to check the important points carefully. It's unlikely, though conceivable, that I missed something that should affect my ratings.

---

> ### Author Rebuttal · Authors · 2023-08-27
>
> We thank all reviewers for their time and for giving valuable and supportive comments. We will take these comments into consideration and make the necessary revisions.
>
> Question 1: Performance gains are relatively small.
>
> We conduct experiments on two document-level and one sentence-level event argument extraction datasets. Compared with multiple baselines, we achieve $1.7\sim5.7$, $2.7\sim21.1$, and $0.5\sim15.6$ point performance improvements on the strict Arg-C F1 metric, respectively. This demonstrates the effectiveness and generalization ability of our method.
>
> Question 2: The long-range trigger-arguments are not fully convincing.
>
> Our method focuses on building intra-event and inter-event dependencies, and to explore whether other dependencies can be captured, we additionally conduct long-range trigger-arguments experiments. Even though our method does not have any modules to capture long-distance dependencies, performance gains are still achieved at multiple distances.
>
> Question 3: Variance of the scores.
>
> We experiment with multiple seeds and report the average performance on the test set. We will add specific experimental details in the appendix.

---

### Official Review · Reviewer_N1pm · 2023-08-05

**Soundness:** 3

**Excitement:**

4: Strong: This paper deepens the understanding of some phenomenon or lowers the barriers to an existing research direction.

**Missing References:**

Lv et al.  2021 (HGEED: Hierarchical graph enhanced event detection)

**Paper Topic And Main Contributions:**

This paper proposed a perspective to automatically construct intra-event and inter-event argument role dependencies, learning the interactions between different roles.

**Questions For The Authors:**

1. The whole structure of the module is not clear in Fig. 2. Only inter and intra event graphs are presented.
2. The inter and intra event graphs are constructed using the trigger and event type information, especially for Inter-event Dependency, why applying similarity between the events, which may not contain useful information.
3. In ablation study, only inter-event dependency is removed, however, in the method, inter and intra are bond with each other, the 1.9 points drop after removing inter-event maynot demonstrate the effect of only inter-event dependency. Why not remove intra-event dependency as well?
4. The idea of construct graphs also shows in Lv et al.  2021 (HGEED: Hierarchical graph enhanced event detection), which constructs sentence-level and document level graphs. Wha's the difference? Or it's just the same idea applying to different tasks.

**Reasons To Accept:**

1. Experimental results on the ACE05, RAMS, and WikiEvents datasets show the great advantages of the proposed approach.
2. Intra-event and inter-event dependency-aware graph network are presented for EAE,  learning rich role semantic information without using manual label templates.

**Reasons To Reject:**

1. The whole structure of the module is not clear in Fig. 2. Only inter and intra event graphs are presented.
2. The inter and intra event graphs are constructed using the trigger and event type information, especially for Inter-event Dependency, why applying similarity between the events, which may not contain useful information.
3. In ablation study, only inter-event dependency is removed, however, in the method, inter and intra are bond with each other, the 1.9 points drop after removing inter-event maynot demonstrate the effect of only inter-event dependency. Why not remove intra-event dependency as well?
4. The idea of construct graphs also shows in Lv et al.  2021 (HGEED: Hierarchical graph enhanced event detection), which constructs sentence-level and document level graphs. Wha's the difference? Or it's just the same idea applying to different tasks.

**Reproducibility:**

4: Could mostly reproduce the results, but there may be some variation because of sample variance or minor variations in their interpretation of the protocol or method.

**Reviewer Confidence:**

5: Positive that my evaluation is correct. I read the paper very carefully and I am very familiar with related work.

---

> ### Author Rebuttal · Authors · 2023-08-27
>
> We thank all reviewers for their time and for giving valuable and supportive comments. We will take these comments into consideration and make the necessary revisions.
>
> Question 1: The whole structure of the module is not clear in Fig.2.
>
> In Fig.2, we show the overall framework of our method. Specifically, we use a specific example to show the whole process, and mark the text description under the specific modules, such as input module, dependency building module, dependency interaction module and argument extraction module. At the same time, we delineate the real similar and dissimilar events in the memory unit. More often, we also use different colors to distinguish argument roles, use solid lines with arrows to indicate the flow of modules, and use dotted lines to indicate dependent interactions. Finally, the specific extraction results of this example are depicted in the extraction module.
>
> Question 2: Why applying similarity between the events, which may not contain useful information.
>
> We apply the similarity between events to retrieve similar events, hoping that the similar event structure and context can provide more information to the current context and help the current extraction. More, our graph assigns different weights to the argument roles, and we have carried out dependency interaction between GCNs in each layer, which can perform information interaction at different granularities and filter redundant information.
>
> Question 3: Why not remove intra-event dependency as well?
>
> Intra-event dependencies are the basis of inter-event dependencies, so we need to first build and store the intra-event dependency graph, and then build the inter-event dependency graph.
>
> Question 4: Same idea Lv et al. 2021 (HGEED: Hierarchical graph enhanced event detection)
>
> HGEED and our method have completely different motivations. HGEED aims to use documents to construct dependency information between sentences, so they build sentence-level and document-level graphs. However, our starting point is to construct the dependency of argument roles within the same event and between similar events, so we construct intra-event and inter-event graphs using event structure as the basic unit, which is not limited to the current context.

---

### Meta-Review · Area_Chair_5A3A · 2023-09-25

**Recommendation:** 3

**Metareview:**

This paper introduces a novel approach for automatically capturing dependencies between event arguments within and across events using graphical networks. Additionally, the incorporation of an event memory and retrieval module during inference adds innovation to the proposed architecture. The experimental results, evaluated on ACE05, RAMS, and WikiEvents datasets, demonstrate the significant advantages of this approach.

Two out of three reviews agree on the good soundness of the work; one reviewer raised concerns over the clarify of the presentation of technical details, which I think is clarified in authors' rebuttal. Overall, I don't see obvious improvement areas in terms of soundness. Reviews also point out that the idea of constructing intra-event and inter-event dependency graphs is novel and well-motivated; however, the performance gains over prior methods appear relatively small based on the results presented.

---

### Decision · Program_Chairs · 2023-10-07

**Decision:**

Accept-Findings

**Comment:**

This paper introduces a novel approach for automatically capturing dependencies between event arguments within and across events using graphical networks. Additionally, the incorporation of an event memory and retrieval module during inference adds innovation to the proposed architecture. The experimental results, evaluated on ACE05, RAMS, and WikiEvents datasets, demonstrate the significant advantages of this approach.

Two out of three reviews agree on the good soundness of the work; one reviewer raised concerns over the clarify of the presentation of technical details, which I think is clarified in authors' rebuttal. Overall, I don't see obvious improvement areas in terms of soundness. Reviews also point out that the idea of constructing intra-event and inter-event dependency graphs is novel and well-motivated; however, the performance gains over prior methods appear relatively small based on the results presented.